# Investigation of gene-gene interactions in cardiac traits and serum fatty acid levels in the LURIC Health Study

**Jiayan Zhou**[1,2], **Kristin Passero**[1,3], **Nicole E. Palmiero**[1], **Bertram Müller-Myhsok**[4,5,6],
**Marcus E. Kleber**[7], **Winfried Maerz**[7,8,9], **Molly A. Hall**[1,3]*

**1** Department of Veterinary and Biomedical Sciences, College of Agricultural Sciences, The Pennsylvania
State University, University Park, PA, United States of America, **2** Department of Statistics, Eberly College of
Science, The Pennsylvania State University, University Park, PA, United States of America, **3** The Huck
Institutes of the Life Science, The Pennsylvania State University, University Park, PA, United States of
America, **4** Statistical Genetics, Max Planck Institute of Psychiatry, Munich, Germany, **5** Munich Cluster of
Systems Biology, SyNergy, Munich, Germany, **6** Institute of Translational Medicine, University of Liverpool,
Liverpool, United Kingdom, **7** Vth Department of Medicine, Medical Faculty Mannheim, Heidelberg
University, Mannheim, Germany, **8** Synlab Holding Deutschland GmbH, SYNLAB Academy, Mannheim and
Augsburg, Germany, **9** Medical University of Graz, Clinical Institute of Medical and Chemical Laboratory
Diagnostics, Graz, Austria

* mah546@psu.edu

doi.org/10.1371/journal.pone.0238304

Hospital and Harvard Medical School, UNITED
STATES

**Data Availability Statement:** Due to the articles of
Ludwigshafen Risk and Cardiovascular Health
(LURIC) Study gGmbH, which needs to

## Abstract

Epistasis analysis elucidates the effects of gene-gene interactions (G×G) between multiple loci for complex traits. However, the large computational demands and the high multiple testing burden impede their discoveries. Here, we illustrate the utilization of two methods, main effect filtering based on individual GWAS results and biological knowledge-based modeling through Biofilter software, to reduce the number of interactions tested among single nucleotide polymorphisms (SNPs) for 15 cardiac-related traits and 14 fatty acids. We performed interaction analyses using the two filtering methods, adjusting for age, sex, body mass index (BMI), waist-hip ratio, and the first three principal components from genetic data, among 2,824 samples from the Ludwigshafen Risk and Cardiovascular (LURIC) Health Study. Using Biofilter, one interaction nearly met Bonferroni significance: an interaction between rs7735781 in *XRCC4* and rs10804247 in *XRCC5* was identified for venous thrombosis with a Bonferroni-adjusted likelihood ratio test (LRT) p: 0.0627. A total of 57 interactions were identified from main effect filtering for the cardiac traits G×G (10) and fatty acids G×G (47) at Bonferroni-adjusted LRT p < 0.05. For cardiac traits, the top interaction involved SNPs rs1383819 in *SNTG1* and rs1493939 (138kb from 5' of *SAMD12*) with Bonferroni-adjusted LRT p: 0.0228 which was significantly associated with history of arterial hypertension. For fatty acids, the top interaction between rs4839193 in *KCND3* and rs10829717 in *LOC107984002* with Bonferroni-adjusted LRT p: 2.28×10$^{-5}$ was associated with 9-trans 12-trans octadecanoic acid, an omega-6 trans fatty acid. The model inflation factor for the interactions under different filtering methods was evaluated from the standard median and the linear regression approach. Here, we applied filtering approaches to identify numerous genetic interactions related to cardiac-related outcomes as potential targets for therapy. The

acknowledge the German Data Protection Act and the consent given by the study participants, data cannot be released to the public domain. The exploitation of the (LURIC) Study database is governed by the articles of the LURIC Study GmbH (non-profit LLC), registered under number HRB 7668 at the commercial registry of Freiburg in Breisgau, Germany. According to the articles of the organization, data access is made available to researchers upon request and approval. This procedure makes sure that rules of good scientific practice are followed and that credit is given to the people who have been in charge of the design and the organization of the study. Interested researchers are invited to address their request or proposal to Kai Grunwald (kai.grunwald@weitnauer.net) or to the Principal Investigator of the LURIC Study, Winfried März (winfried.maerz@synlab.com) who are in charge of supervising ethical and legal aspects of the LURIC study. Finally, the authors confirm that they accessed these data upon approval by LURIC and that all other researchers can access the data upon approval in the same manner the authors did.

**Funding:** This work is supported by the USDA National Institute of Food and Agriculture and Hatch Appropriations under Project #PEN04275 and Accession #1018544. The funders had no role in study design, data collection and analysis, decision to publish, or preparation of the manuscript. Genotyping of the LURIC study participants was supported by the 7th Framework Program AtheroRemo (grant agreement #201668) of the European Union. The funders had no role in study design, data collection and analysis, decision to publish, or preparation of the manuscript. The Synlab Holding Deutschland GmbH provided support in the form of the salaries for author W.Z. and M.E.K. but did not have any additional role in the study design, data collection and analysis, decision to publish, or preparation of the manuscript. The specific roles of these authors are articulated in the 'author contributions' section.

**Competing interests:** The authors have read the journal's policy and have the following competing interests: WZ and MEK are paid employees of The Synlab Holding Deutschland GmbH. There are no patents, products in development or marketed products associated with this research to declare. This does not alter our adherence to PLOS ONE policies on sharing data and materials.

approaches described offer ways to detect epistasis in the complex traits and to improve precision medicine capability.

## Introduction

As sequencing technology improves, the understanding of genetic variation and their interactions is in high demand to interpret the trigger of complex diseases. Computational approaches to identify epistasis take advantage of improved sequencing technology [1]. Previously, multiple gene-gene interactions G×Gs have been identified in cardiovascular diseases [2], atrial fibrillation [3], dyslipidemia [4], type 2 diabetes (T2D) [5], and cancers [6, 7] from main effect filtering based on the genome-wide association studies (GWAS) and/or the analysis of specific candidate gene-gene pairs. As evidenced, applying high throughput methods for detecting G×Gs allows for novel discoveries. However, the heavy computational demands on pairwise single nucleotide polymorphism (SNP) models restrain the identification of these interactions [2]. Hence, there is a need to reduce the number of tests through filtration or pre-selection of candidate SNPs in order to identify SNP-SNP interactions.

Main effect filtering is a widely used and simple way to reduce the multiple testing burden by selecting suggestive SNPs from GWAS outputs and using them to generate pairwise SNP models for G×G analysis. An alternative method is candidate SNP/gene selection where plausible genetic variants are selected based on relationships between genes published in literature and/or from databases. Biofilter software is a tool which provides a convenient single interface for accessing multiple publicly available human genetics data resources through the Library of Knowledge Integration (LOKI) [8]. It is used to match SNPs to genes and ascertain the known biological relationship between these genes, including physical interactions of the encoded proteins, pathways, and ontological categories, leveraging them to model putative genetic interactions.

The Ludwigshafen risk and cardiovascular (LURIC) health study is an ongoing study to explore risk factors for cardiovascular diseases and is aimed at the understanding of genes, environments, medications, and their associations with cardiac health [9]. Data were collected from questionnaires, angiograms, and laboratory examinations with several follow-ups to capture information on longitudinal health outcomes. LURIC aims to investigate common cardiac traits, such as coronary artery disease (CAD), arterial hypertension, T2D, myocardial infarction (MI), stroke, peripheral vascular disease (PVD), and atherosclerosis, and potential environmental risk factors, including lipids, medications, and chemical exposures.

Alongside the use of the main effect filtering from GWAS, we suggest the exploitation of biological knowledge from different databases to generate SNP pairs among genes to limit the number of tests [10]. Many G×Gs were identified from main effect filtering and one knowledge-based interaction was identified based on Biofilter modeling. Using these filtering methods to inform selection of gene-gene models will allow the detection of interacting genetic risk factors in complex diseases.

## Materials and methods

### Ludwigshafen risk and cardiovascular (LURIC) health study

The LURIC health study is a prospective cohort study aimed at assessing the genetic and environmental risk factors in cardiovascular diseases for 3,316 Caucasians of German ancestry (2,309 male and 1,007 female) between 1997 and 2000 [9]. Participants with acute illness

(other than the acute coronary syndromes), non-cardiac chronic diseases, and/or malignant neoplasms in the past five years were excluded. The study was approved by the Landesärzte-kammer Rheinland-Pfalz ethics committee and was restricted by the Declaration of Helsinki with appropriated written informed consent for every participant.

## Genotyping and SNP selection

687,262 SNPs were genotyped for 3,061 samples using the Affymetrix Genome-Wide Human SNP Array 6.0 from peripheral blood samples [9, 11, 12]. Quality control (QC) for genotypes was performed in PLINK 1.90 software (Fig 1) [13]. Participants were filtered to remove individuals below the age of 18; persons with missing covariates including age, waist-hip ratio, BMI, and sex; and samples with less than 99% sample call rate. The genetic and reported sex for individuals were compared through a sex concordance check. SNPs were dropped if they had a less than 99% genotype call rate and/or a less than 5% for minor allele frequency (MAF) and were further LD pruned with an $r^2$ threshold of 0.75. Identity by descent (IBD) was calculated between all pairs of samples to assess for relatedness (kinship coefficient > 0.125). Individuals with higher missingness were dropped within related pairs. To ensure we retained > 99% sample and genotype call rates and > 5% MAF following the sample drop based on IBD, we performed a second round of QC. Principal components were calculated based on the EIGENSTRAT method [14]. After the genotype quality-control, 2,824 samples and 577,007 SNPs remained for analysis.

## Phenotype definitions, disease classifications, and laboratory measurements

In this study, we aimed at identifying interactions between SNPs associated with cardiac traits based on clinical laboratory measurements, and/or questionnaires. The methods for the classification of the CAD, PVD, cardiac-related diseases and valve diseases were described previously [9]. Specifically, CAD patients were further classified based on the number of visible luminal narrowing ($\geq$ 20% stenosis) in 15 coronary segments. Diabetes mellitus cases were determined by the level of plasma glucose ($\geq$ 126 mg/dl) or 2-hour plasma glucose concentration ($\geq$ 200 mg/dl after oral intake of 75 g glucose) including considering the level of HbA1c ($\geq$ 6.5) as per the guidelines of the American Diabetes Association (ADA) and the World Health Organization (WHO) [9, 11]. Persons with a history of diabetes mellitus were also classified as diabetic (type I and type II diabetes). The ambiguous cases were considered as T2D, since the categories of diabetes could not be identified based on the age and the use of insulin treatment. Hypertension was determined based on blood pressure exceeding 140/90 mmHg and a history of hypertension [9, 17]. Demographic information was gathered by questionnaires or physical examinations. The erythrocyte fatty acid composition was measured through the HS-Omega-3 Index® methodology as previously described [18, 19]. The amount of individual fatty acid was reported as the percentage of total identified fatty acids after corrected with response factor.

## Quality control for phenotypes

Thirty-three cardiac traits and 27 fatty acids for 3,316 samples were selected. Quality control for cardiac traits and fatty acids were mainly performed with the CLARITE R package (https://github.com/HallLab/clarite) (Fig 1) [15]. Participants were filtered to remove children/adolescents and those with missing covariates. Variables with a case or control size less than 150 were dropped to ensure the sufficient sample pool and to keep as many variables as possible. The same protocol was followed for fatty acids, except the minimum sample size was set as

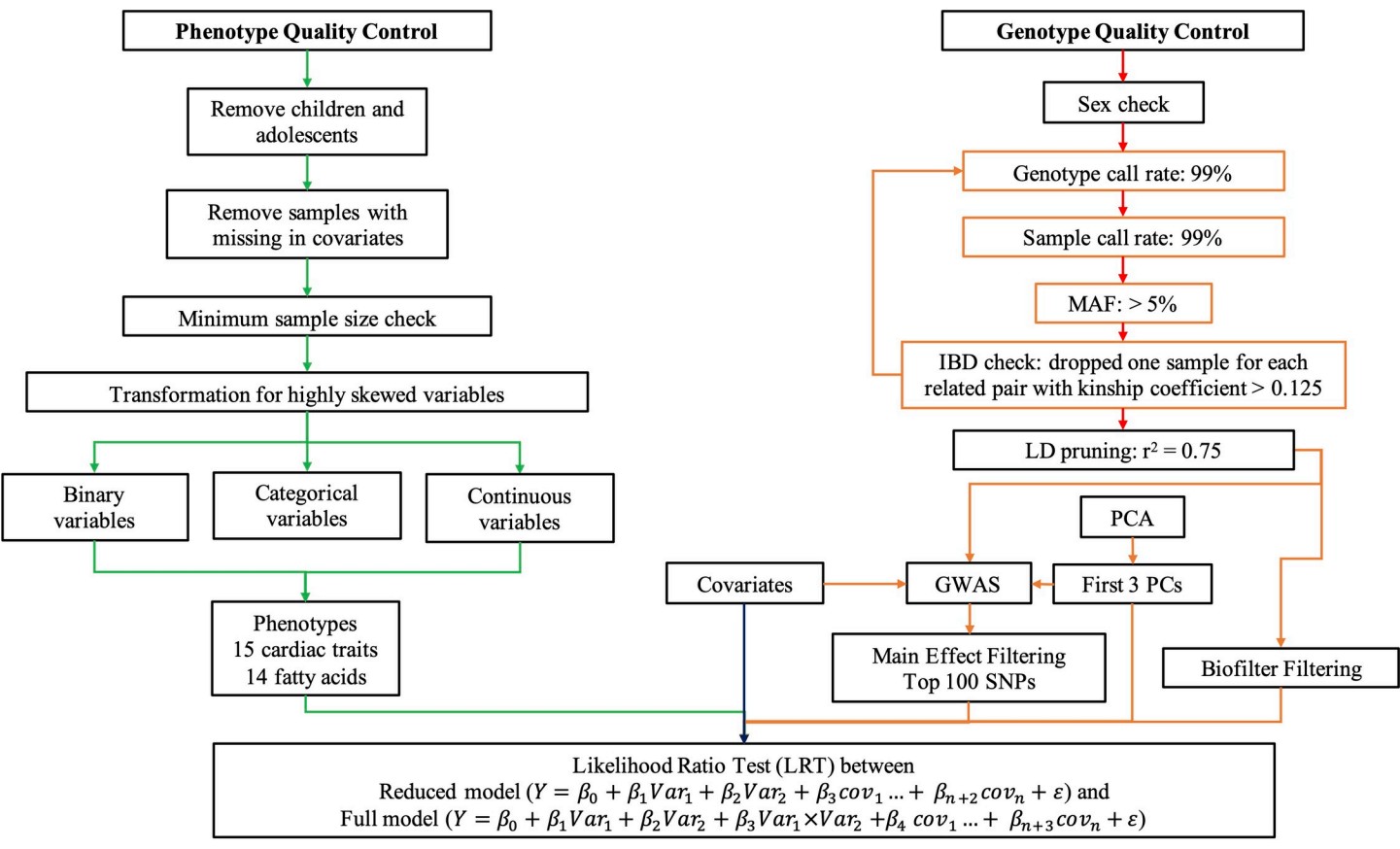

**Fig 1. Design of quality control and gene-gene interaction analysis.** The phenotype quality control was performed in CLARITE software [15]. The genotype quality control and the principle component analysis (PCA) were performed in PLINK 1.90 [13]. The genome-wide association study (GWAS) and gene-gene interaction (G×G) were performed in PLATO [16]. SNP-SNP interaction models were assessed using a likelihood ratio test (LRT) between the reduced model and full model, where *Var1* and *Var2* are the potential SNP predictors of interest, *Y* is the phenotype, *cov*s are the covariates, and $\varepsilon$ is the error.

n = 200 as recommended [20, 21] and variables with skewness larger than 0.5 underwent ln(x +1) transformations. The remaining variables (32) were categorized into binary, categorical, or continuous. Only the binary and continuous outcomes (29) were kept for GWAS and G×G analysis including 15 cardiac traits for vary samples and 14 fatty acids for 2,824 samples remained (S1 Table).

## Interaction analysis

To reduce the high multiple testing burden associated with running pairwise genetic interactions between all genome-wide SNPs, two filtering methods were used for selecting the SNPs based on 1) the main effect from individual GWAS results (S1 Fig B-GGG and S1 File) and 2) building SNP pair models based on biological knowledge. All GWAS and G×G were performed using PLATO [16], a software that was designed to provide a standard platform to study complex associations, such as G×G, gene-environment interaction (G×E), environment-wide association study (EWAS) and phenome-wide association studies (PheWAS). All G×G models were run assuming an additive genetic model and adjusting for age, sex, BMI, waist-hip ratio based on CDC report recommendations [22] and additional ANOVA tests (S2 Table), as well as the first three principal components which explained 0.23% of genetic variation (S2A and S2B Fig). SNP-SNP interaction models were assessed using a likelihood ratio test (LRT) between the

reduced model ($Y = \beta_0 + \beta_1 Var_1 + \beta_2 Var_2 + \beta_3 cov_1 \ldots + \beta_{n+2} cov_n + \varepsilon$) and full model ($Y = \beta_0 + \beta_1 Var_1 + \beta_2 Var_2 + \beta_3 Var_1 \times Var_2 + \beta_4 cov_1 \ldots + \beta_{n+3} cov_n + \varepsilon$), where *Var1* and *Var2* are the potential SNP predictors of interest, *Y* is the phenotype, *cov*s are the covariates, and $\varepsilon$ is the error. This test evaluates the significance of the interaction between *Var1* and *Var2* and its contribution above and beyond the main effects of the two individual variates combined.

**Main effect filtering and Biofilter filtering.** Individual GWAS for each cardiac trait and fatty acid phenotype were performed (29 GWAS total) (S1 Fig and S1 File). The top one hundred SNPs were selected from each phenotype's GWAS based on power estimations with 1000 replications in R software using logistic regression model for cardiac traits and linear regression model for fatty acids (S3A and S3B Fig). Power was also estimated under different sample sizes (S3C and S3D Fig) with statistical simulations to ensure the power of our tests. 4,950 of pairwise combinations of each phenotype's top SNPs were assessed for interaction as the main effect filtering. All SNPs passing the genotype quality control were imported into Biofilter software to generate SNP-SNP interaction pairs [8, 10, 23]. Biofilter software builds SNP-SNP pairs by referencing publicly available biological knowledge, including protein-protein interactions, pathways, and ontological information. Biofilter software has been successfully applied to numerous phenotypes, including lipids [10, 24–26], efavirenz-containing HIV treatment [27], multiple-sclerosis [28], and cataracts [29, 30], to identify genetic interactions through knowledge-based filtering, allowing the possibility of identifying SNPs that interact without demonstrating a main effect. 19,585 of SNP pairs were modeled for all phenotypes based on previous knowledge which has been identified with at least five database sources by Biofilter using a gene boundary of 5kb.

**Model inflation checks and visualization.** To compare the performance of models, model inflation factors were calculated and visualized through the Quantile-Quantile (Q-Q) plots. A series of referential G×G was performed with the same set of 1000 random selected SNPs (499,500 SNP-SNP models) as the reference for comparing the performance of the filtering methods in each variable. A quantile function *qchisq* of the Chi-square distribution was applied for evaluating the model inflation from the p-values based on the standard median approach and a linear regression approach between the observed values and expected values on R software for G×Gs under each phenotype (S3 Table). The distributions of the model inflation from different approaches were plotted. Q-Q plots were created for the referential G×G and G×G analyses with two filtering methods for each variable by using *qqman* package in R software.

Volcano-like plots were generated from R software for elucidating the associations between the significance and estimated interaction coefficient (-log10 (LRT p-value) against interaction beta-coefficient) for interactions, highlighting interactions with significant adjusted LRT p-values. The distributions of the phenotypes and genotypes for significant interactions were plotted by using *ggplot2* package in R software. The genotypes of the SNPs in top interaction were plotted for the cardiac traits by counting the number of case/control with using the *plot3D* package in R software.

The SNPs involved in interactions with the false discovery rate (FDR) less than 0.05 were further annotated with gene by Biofilter using a gene boundary of 5kb. These significant interactions were selected for network plots if both SNPs were located within a gene or locus. Two network plots were generated using *igraph* package in R software at gene level based on the significant gene-gene interaction with (1) Bonferroni-adjusted p < 0.05 and (2) FDR less than 0.05.

## Results

### Interaction based on Biofilter and main effect filtering

For the cardiac G×G, one SNP pair was identified through Biofilter modeling associated with venous thrombosis or pulmonary embolism: rs7735781 in *XRCC4* and rs10804247 in *XRCC5* (uncorrected LRT p = $3.20 \times 10^{-6}$ and Bonferroni-adjusted LRT p = 0.0626996) (S4 Table). The main effect filtering approach yielded 10 models with a Bonferroni-adjusted LRT p-value less than 0.05, including four SNP-SNP models associated with venous thrombosis or pulmonary embolism and six SNP-SNP models associated with history of arterial hypertension (S4 Table). The top results involved SNPs rs11642027 (intergenic) and rs6986305 in *LOC105375746* (Bonferroni-adjusted LRT p: 0.0104) associated with venous thrombosis or pulmonary embolism. Other top results included SNPs rs1493939 (intergenic) and rs1383819 in *SNTG1* (Bonferroni-adjusted LRT p: 0.0228), rs1389542 (intergenic) and rs1383819 in *SNTG1* (Bonferroni-adjusted LRT p: 0.0280), and rs1493939 and rs12547263 in *SNTG1* (Bonferroni-adjusted LRT p: 0.0286) all for history of arterial hypertension.

For the fatty acids G×G, the main effect filtering approach yielded 47 models with a Bonferroni-adjusted LRT p-value less than 0.05, including 7 models for log-transformed 9-cis 12-trans octadecanoic acid, 38 models for log-transformed 9-trans 12-trans octadecanoic acid, and 2 models of SNPs for log-transformed trans-palmitoleic acid (S4 Table). The top results for log-transformed 9-cis 12-trans octadecanoic acid involved SNPs rs6129232 (16kb 5' of *AL121588.1*) and rs4923022 in *GAS2* (Bonferroni-adjusted LRT p: 0.0152, rs6129232 and rs337454 in *GAS2* (Bonferroni-adjusted LRT p: 0.0296), and rs6129232 and rs337459 in *GAS2* (Bonferroni-adjusted LRT p: 0.0348). The first two significant interactions for log-transformed 9-trans 12-trans octadecanoic acid involved SNPs rs10829717 (intergenic) and rs4839193 in *KCND3* (Bonferroni-adjusted LRT p: $2.280 \times 10^{-5}$), and rs10829717 and rs11102365 in *KCND3* (Bonferroni-adjusted LRT p: $2.640 \times 10^{-5}$). The top significant result for log-transformed trans-palmitoleic acid involved SNPs rs2227016 in *HUU95743* and rs351219 in *STRA6* (Bonferroni-adjusted LRT p: 0.0078). Another significant interaction for log-transformed trans-palmitoleic acid involved SNPs rs12547615 (intergenic) and rs875979 in *GALNT1* (Bonferroni-adjusted LRT p: 0.0453).

### Distribution of LRT p-values and model inflation

To check for inflation of results, the distribution of LRT p-values for the referential G×Gs as the baseline and G×Gs under two filtering methods was plotted for each variable (Figs 2 and S4). The model inflations for 9 of 29 phenotypes were higher or lower than expected (Fig 2). The corresponding genomic inflation factors were calculated and plotted for each SNP-SNP model (Fig 3 and S3 Table). Among the cardiac traits, the observed LRT p-values for the referential G×Gs were similar to the expected LRT p-values under different phenotypes. Notably, most of the estimated LRT p-values for G×Gs under Biofilter modeling were slightly underestimated except for the smallest estimations in venous thrombosis or pulmonary embolism and T2D. However, the estimations of small LRT p-values under main effect filtering varied, with obvious underestimations in stroke and overestimations in atrial fibrillation, the history of arterial hypertension, and venous thrombosis or pulmonary embolism. Within the fatty acid results, the distributions of small LRT p-values for the referential G×Gs were diverse with slight differences for each fatty acid. However, the estimations of LRT p-values on interactions based on main effect filtering were overestimated for log-transformed 9-cis 12-trans octadecanoic acid, log-transformed 9-trans 12-trans octadecanoic acid, and log-transformed trans-palmitoleic acid.

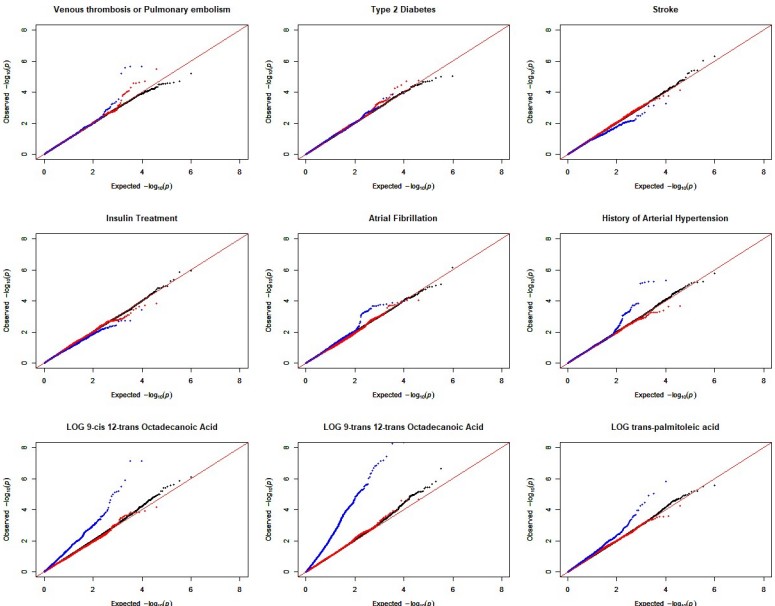

**Fig 2. Quantile-quantile (Q-Q) plots of observed LRT p-values and referential G×Gs using two filtering methods.**
G×G based on 1000 random selected SNPs was performed as a baseline to assess the inflation (Black). G×G under
main effect filtering (Blue) and G×G under Biofilter filtering (Red) were performed and compared with the G×G based
on 1000 random selected SNPs for each phenotype. Individual Q-Q plots were generated to visualize the observed LRT
p-values to the expected LRT p-values for cardiac traits and fatty acids. The red line represents the ideal estimation of
LRT p-values. The corresponding genomic inflation factors were calculated for showing the model inflation (S3 Table).

## Relationship between LRT p-value and beta coefficient of interactions

Volcano-like plots were created based on the interaction beta coefficient in the full model and
uncorrected LRT p-value for interactions in cardiac traits and fatty acids under two filtering
methods (Fig 4). Several significant interactions for fatty acids were identified under main
effect filtering. However, the fatty acids tended to demonstrate low effect sizes with beta coeffi-
cients close to zero. Generally, larger effect sizes were found for genetic interactions identified
with Biofilter rather than main effect filtering.

## Distribution of phenotypes and genotypes

Plots of the phenotypic distribution alongside the corresponding genotypes were generated for
visualizing the most significant interaction of each phenotype (Figs 5 and 6). The phenotypes
and genotypes for 100 randomly selected individuals were also plotted for the significant inter-
actions (Figs 5B, 5D, 5F and 6B, 6D and 6F). For cardiac traits, the distributions of genotypes
between individuals with or without the cardiac traits are comparable. The breakdown of the
number of cases/controls by genotype combination are displayed for the top results: venous
thrombosis/pulmonary embolism using main effect filtering (Fig 7) and biofilter (Fig 8) and
hypertension (Fig 9) using main effect filtering. For example, in venous thrombosis/pulmo-
nary embolism for the top SNP-SNP interactions identified with main effect filtering, the
genotype combination with the largest number of cases was under the combination of GG for
rs6986305 and CT for rs11642027 with 38 cases. Conversely, the genotype combination with
the largest number of controls was CG for rs6986305 and CT for rs11642027 with 580 controls
(Fig 7). The variation of genotype for the interaction (rs10804247-rs7735781) under Biofilter
filtering and modeling is more conservative than the identified interactions under main effect

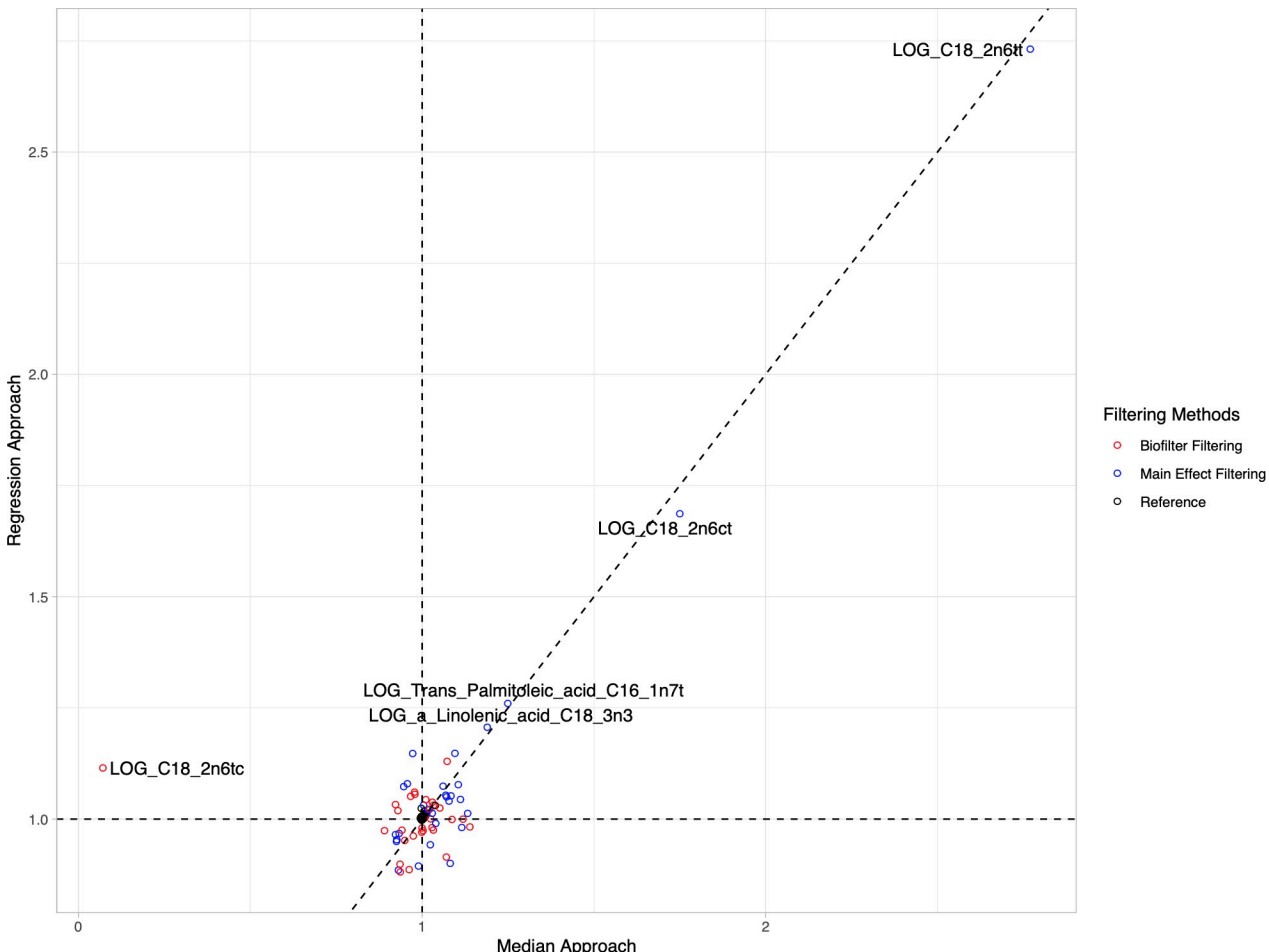

**Fig 3. Comparison of model inflation based on genomic inflation factor among different approaches.** The model inflation for each interaction was calculated (S3 Table) using the median approach and regression approach for different filtering methods (Red: Biofilter filtering, Blue: Main effect filtering, and Black: Reference). The horizontal and vertical black dotted lines represent the inflation factor at 1 for different approaches. The line of equality was also plotted for viewing the estimation differences from models. Five models with comparably large inflation were labeled.

filtering. For fatty acids, the distributions of genotypes for each phenotype were diverse and there was no conserved pattern for the individuals with different concentrations of fatty acids. For the top interaction for log-transformed 9-cis 12-trans octadecanoic acid under main effect filtering, the majority of genotypes are GG for rs10989148 and TC for rs11711981, but with a significant portion of CC alleles for rs11711981 as well (Fig 6A). For the top interaction for 9-trans 12-trans octadecanoic acid under main effect filtering, the majority of samples are TT for rs10829717 and AA for rs4839193, but many CA genotypes also appeared for rs4839193 (Fig 6C). For the top interaction for log-transformed trans palmitoleic acid (C16:1n7t) under main effect filtering, the majority of genotypes are TT for rs2227016 with approximate equal distribution of CT and TT for rs351219 (Fig 6E).

## Mapping significant interactions at the gene level

For a global representation of the results, two interaction maps were plotted for the Bonferroni significant at 0.05 (Fig 10) and FDR less than 0.05 (Fig 11) SNP-SNP interaction results. This network representation allowed us to identify genes involved in multiple interaction: putative

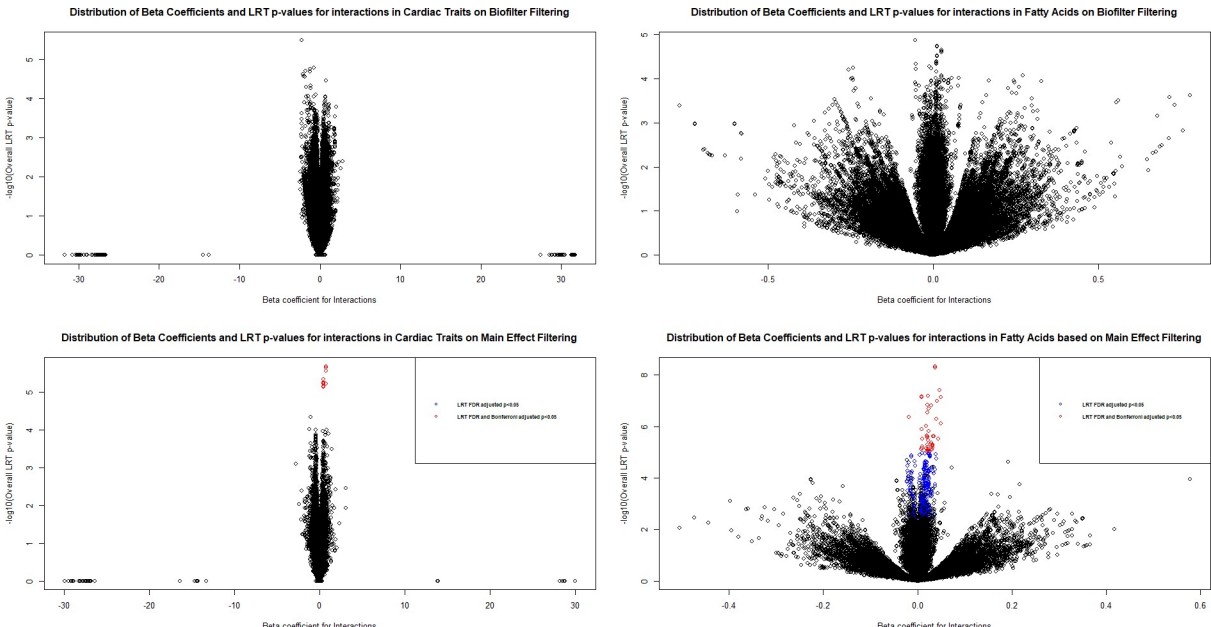

**Fig 4. Distribution of interaction beta coefficients and LRT p-values for interactions in cardiac traits and fatty acids under different filtering methods.** The beta coefficients and LRT p-values for interactions are plotted by -log$_{10}$(LRT p) against interaction beta coefficient. The significant interactions are highlighted according to the significance level of LRT p-values (blue: FDR-adjusted LRT < 0.05; red: Bonferroni-adjusted LRT p < 0.05).

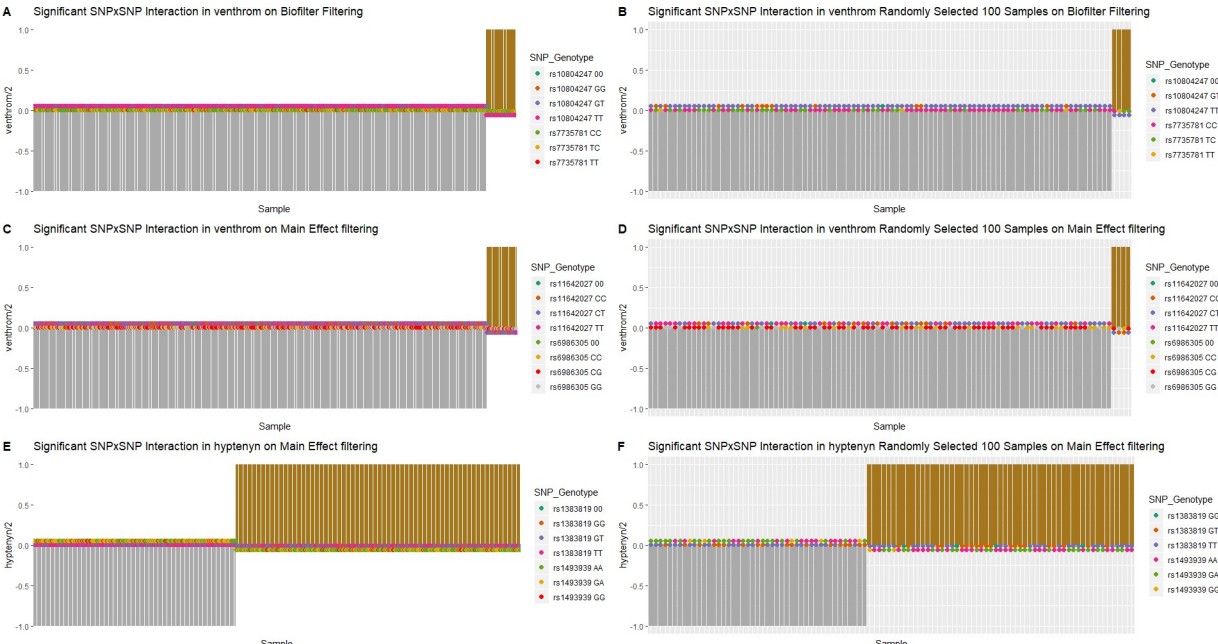

**Fig 5. Distribution of phenotype and genotype for the top interactions in cardiac traits under different filtering methods.** Individuals with cardiac trait were assigned "1" and individuals without cardiac trait were assigned as "-1". The distribution of cardiac trait status and corresponding genotype for SNPs for the top interactions were plotted for A) rs10804247-rs7735781 for the patients with venous thrombosis or pulmonary embolism under Biofilter filtering, C) rs11642027-rs6986305 for the patients with venous thrombosis or pulmonary embolism under main effect filtering, and E) rs1383819-rs1493939 in for the patients with the history of hypertension under main effect filtering. The randomly selected 100 individuals were also plotted for each top interaction (B, D, F).

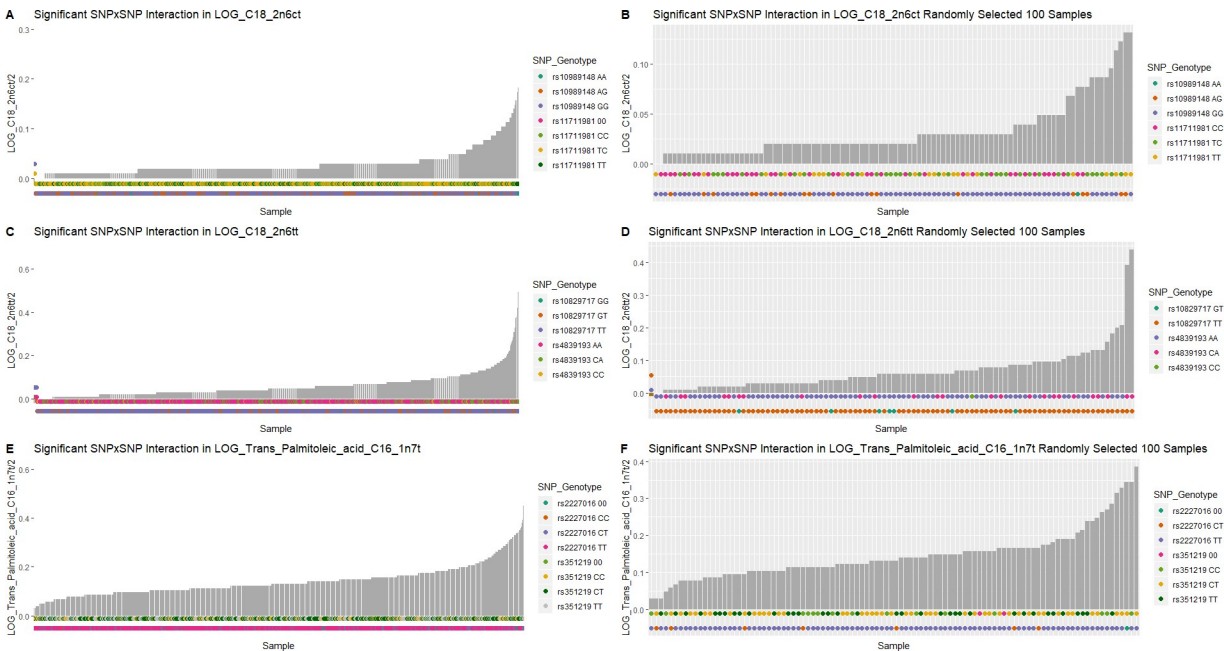

**Fig 6. Distribution of phenotype and genotype for the top interactions in fatty acids under different filtering methods.** The distribution of fatty acid concentration and corresponding SNP genotype for SNPs for the top interactions were plotted for A) rs10989148-rs11711981 in log-transformed 9-cis 12-trans octadecanoic acid under main effect filtering, C) rs10829717-rs4839193 in log-transformed 9-trans 12-trans octadecanoic acid under main effect filtering, and E) rs2227016-rs351219 in log-transformed trans palmitoleic acid (C16:1n7t) under main effect filtering. The randomly selected 100 individuals were also plotted for each top interaction (B, D, F).

"gene hubs". Only the interactions in which both SNPs mapped to a gene (within a 5kb upstream/downstream gene boundary) were considered. Results that were mapped to a gene

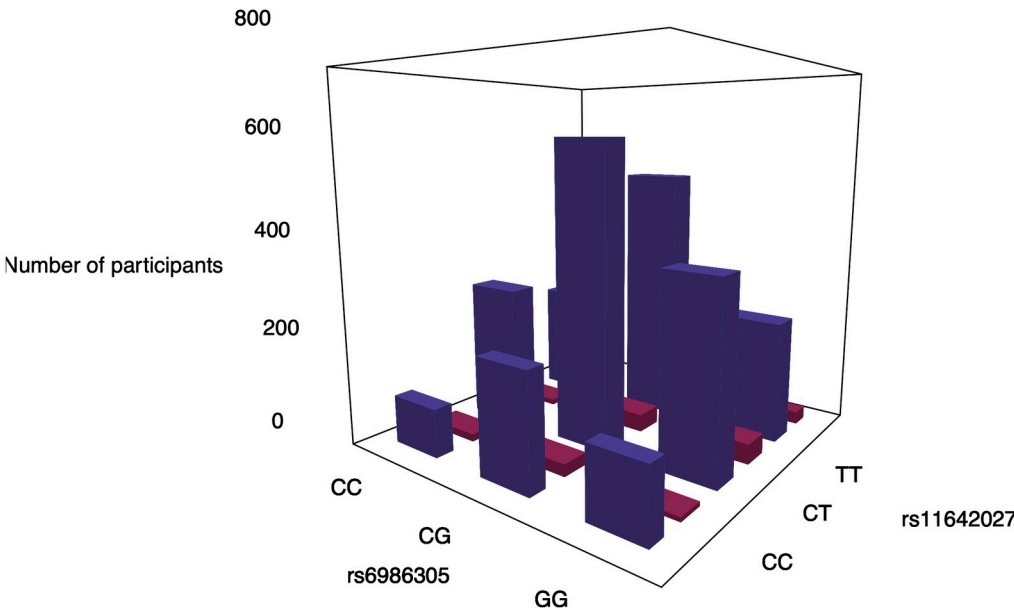

**Fig 7. Genotype combinations between rs6986305 and rs11642027 for the participates with venous thrombosis or pulmonary embolism under main effect filtering.** The genotypes of each SNPs were plotted with the number of the participants with venous thrombosis or pulmonary embolism (cases: red) or not (controls: purple).

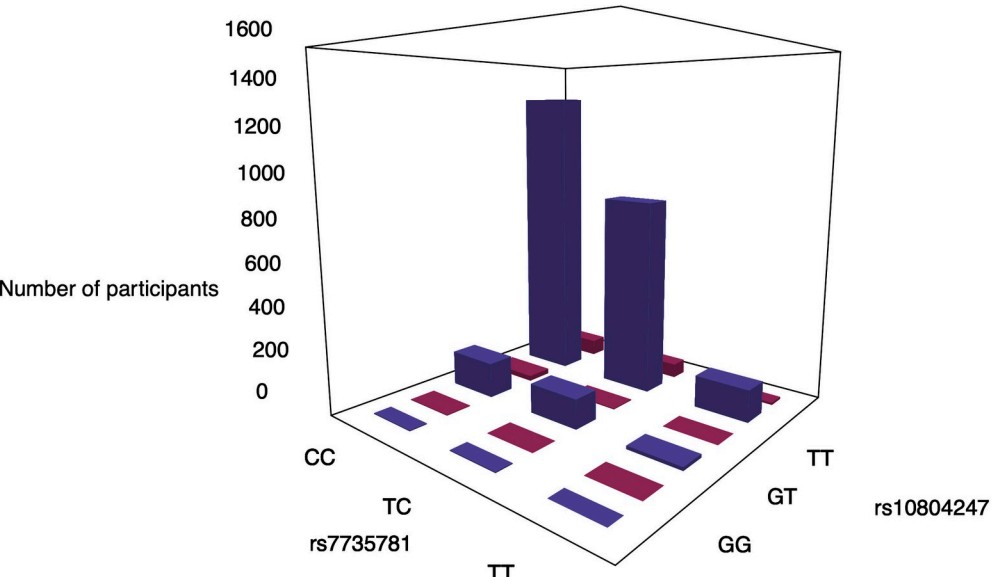

**Fig 8. Genotypes combinations between rs7735781 and rs10804247 for the participates with venous thrombosis or pulmonary embolism under Biofilter filtering.** The genotypes of each SNPs were plotted with the number of the participants with venous thrombosis or pulmonary embolism (cases: red) or not (controls: purple).

and also met a Bonferroni significance threshold with alpha of 0.05 were only found in one phenotype: 9-trans 12-trans octadecanoic acid, an omega-6 fatty acid (C18:2n6tt). Here, two genes, *GCNT1* and *SCN4B*, were involved in multiple significant interactions (Fig 11). *GCNT1* was found to interact with *SCN2B*, *SCN4B*, *ASTN2*, and *LOC105378332*, while *SCN4B* was found to interact with *TMEM135*. To allow for a less conservative significance threshold for our network mapping, we evaluated results with LRT FDR less than 0.05. For this assessment,

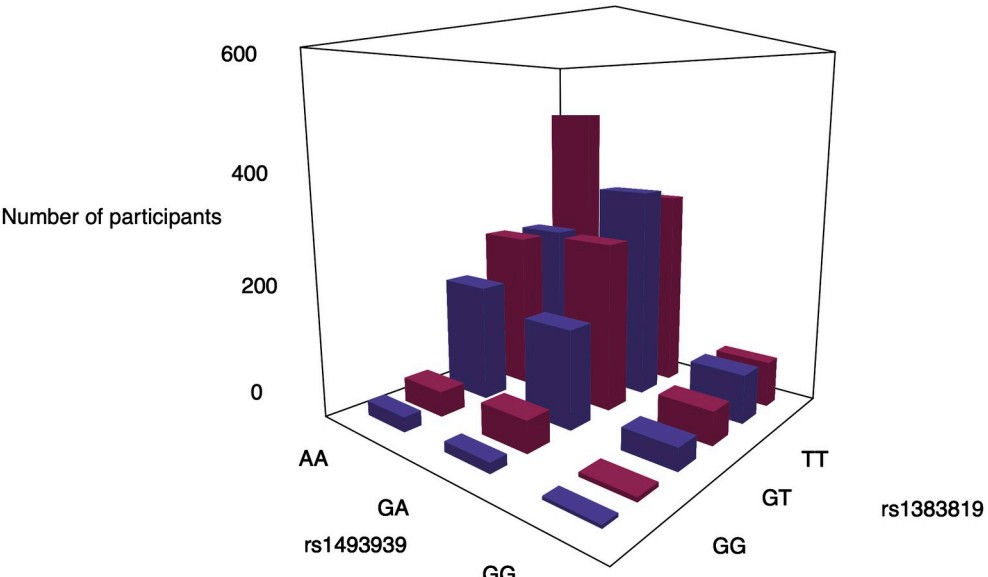

**Fig 9. Genotypes combinations between rs1383819 and rs1493939 for the participates with hypertension under main effect filtering.** The genotypes of each SNPs were plotted with the number of the participants with hypertension (cases: red) or not (controls: purple).

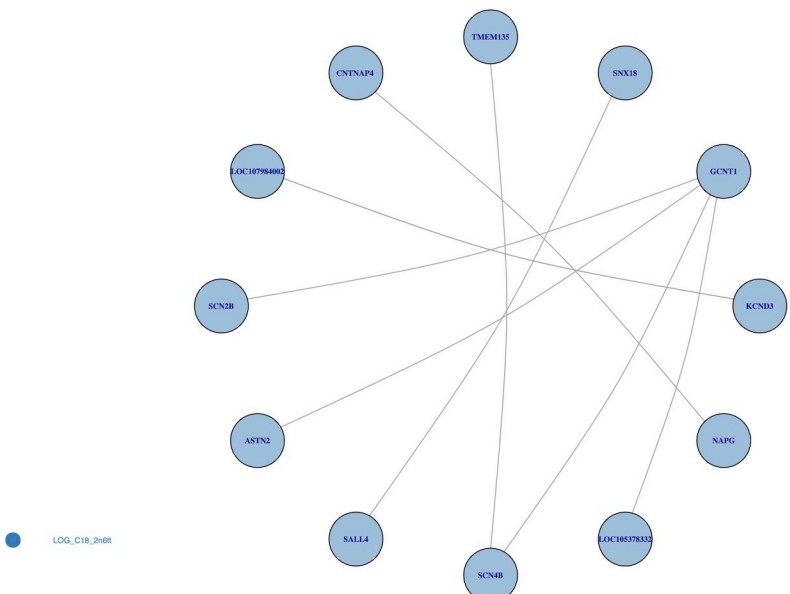

**Fig 10. Mapping significant interactions at the gene level with Bonferroni adjusted p < 0.05.** The Bonferroni significant interactions with gene annotations were selected. Only the omega-6 fatty acid (C18:2n6tt) had Bonferroni significant interactions that mapped to genes. Genes *GCNT1* and *SCN4B* were involved in multiple interactions.

we found that many genes were involved in multiple gene-gene interactions, such as *GCNT1*, *SCN2B*, *NRG2*, and *DNAH5* for omega-6 fatty acid (C18:2n6tt), *PTPRD* for omega-6 fatty acid (C18:2n6ct), and *LCP1* for trans palmitoleic acid (C16:1n7t).

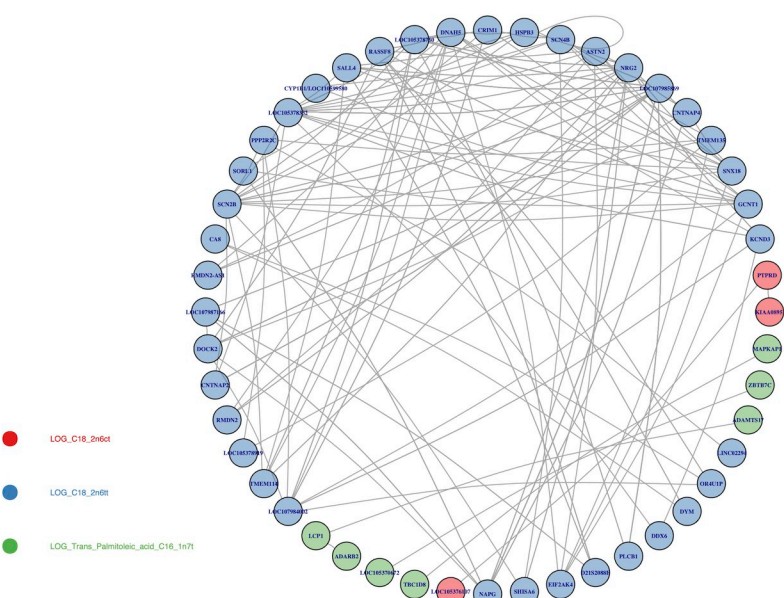

**Fig 11. Mapping significant interactions at the gene level with FDR less than 0.05.** The significant interactions with FDR less than 0.05 with appropriate gene annotations were selected. Two omega-6 fatty acids (C18:2n6ct in red and C18:2n6ct in blue) and the trans-palmitoleic acid (C16:1n7t in green) had significant interactions that mapped to genes. Multiple genes, including *DNAH5* and *NRG2*, were involved in more than one interaction.

## Discussion

The large demands of computational resources and the multiple testing burden impact our ability to find G×G from genome-wide analyses. In this study, we utilized two filtering methods to reduce the computational impediments and identified a number of significant interactions for cardiac traits and fatty acids in the LURIC Health Study. We compared results when filtering using Biofilter with main effect filtering. In this genetic interaction analysis of cardiac health, we identified 57 SNP-SNP models that were Bonferroni significant using a main effect filter and one SNP-SNP model nearly the Bonferroni significant using Biofilter software.

Multiple genes involved in the identified interactions from this study have previously reported associations with cardiac traits. One SNP-SNP model in genes *XRCC4* and *XRCC5* was generated by Biofilter and identified as a significant interaction for venous thrombosis and/or pulmonary embolism. The proteins expressed by *XRCC4* and *XRCC5* function together as a DNA ligase complex in facilitating the reparation of double-stranded DNA breaks through non-homologous end joining [31, 32] and in the completion of V(D)J recombination in the development of the lymphocytes [33, 34]. *XRCC5* is associated with the development of the thalidomide-related thrombosis as the response to the red blood cell apoptosis, which may associate to the formation of venous thrombosis and the development of CAD, PVD, stroke, and other cardiac diseases [35–37]. The genetic polymorphism of *XRCC4* has been frequently reported in liver cancer [38] and glioma [39].

A series of *SNTG1*-involved G×Gs were significantly associated with history of arterial hypertension from main effect filtering. The gene *SNTG1* encodes a member protein of the syntrophin family and plays an important role in mediating gamma-enolase trafficking and neurotrophic activity [40]. Mutations in *SNTG1* have been found to lead to idiopathic scoliosis [41] and increase the systolic blood pressure which may contribute to hypertension [42]. *KCND3*-involved G×G were identified for the log-transformed 9-trans 12-trans octadecanoic acid from main effect filtering. *KCND3* is a part of a voltage-gated potassium channel which is widely expressed within the body and controls neurotransmitter release, heart rate, muscle contractions, and other cellular functions [43]. It is associated with body mass index [44], visceral adipose [45], atrial fibrillation [46–49], cardiac repolarization [49], and heart hypertrophy [50]. The interaction between SNPs in *KCND3* and *LOC107984002* on 9-trans 12-trans octadecanoic acid may imply an association between omega-6 trans fatty acid, body fat deposition, and cardiac disease. These disease-related genes and their involved interactions may contribute to elucidating genetic risk factors with fatty acids and cardiovascular diseases. Further validation may explain the biological implications of our significant findings for G×G in various cardiac diseases.

Beyond the individual significant gene-gene interactions, many genes were involved in numerous identified interactions in this study. *GCNT1* was found in this study to interact with four genes (*SCN2B*, *SCN4B*, *ASTN2*, and *LOC105378332*) in association with 9-trans 12-trans octadecanoic acid, an omega-6 fatty acid. Notably, high expression of the *GCNT1* has been previously reported in multiple cancers, including in prostate cancer [51–55]. Consumption of omega-6 fatty acids has been shown to increase risk of developing prostate cancer from randomized trials [56], human cell experiment [57], and mice model [58]. Two other genes that were found in this study to interact in association with *GCNT1* are *SCN2B* and *SCN4B*, components of voltage-gated sodium channels, and both of these factors have been previously associated with the developing prostate cancer [59, 60] and promoting cancer metastasis [61]. The results of our study indicate that *GCNT1* and its interacting factors may function via fatty acid related pathways, which promote the development of prostate cancer among people who intake higher amount of omega-6 fatty acids. Besides prostate cancer, gene *GCNT1* has previously been linked to neural phenotypes, including the cranial width [62] and cognitive

measurements [63]. *ASTN2*, another gene which interacts with *GCNT1*, also has demonstrated associations with neural phenotypes including attention-deficit/hyperactivity disorder (ADHD) [64, 65], autism spectrum disorders (ASD) [65], neuronal development [65, 66], and Alzheimer's disease [67]. The interaction between *GCNT1* and *ASTN2* may affect the level of omega-6 fatty acids, leading to a unbalanced ratio between the omega-3 and omega-6, and then influences the brain functions and cognitions indirectly [68, 69]. Furthermore, *SCN2B* and *SCN4B* were also identified in two studies for their potential effects in the neuronal development, brain functions and cognitive diseases [70, 71]. These results demonstrate how considering multiple interactions can lead to novel hypotheses for further investigation.

The model inflations were estimated from the lambda calculations and visualized from the QQ plots for each filtering method. Estimations of LRT p-values for interactions based on Biofilter modeling were found to be more conservative without significant under- or over-estimations when compared with the main effect filtering. However, using the main effect filtering approach would allow more Bonferroni and FDR significant G×G findings. The majority of the main effect-identified G×Gs demonstrated smaller effects on the phenotypes than the Biofilter-generated models. Upon further investigation of previously published work involving both biofilter and main effect filtering for gene-gene interaction analysis, other examples show that the biofilter approach yielded larger interaction effect sizes than main effect filtering, for example, the gene-gene interaction analyses of lipids [25] and type II diabetes [16].

The discovery of interactions was restricted by a relatively small sample size, potential for asymptomatic individuals, small variation in age range, and lack of ethnic diversity [9]. An additional limitation faced is that knowledge-based G×G discovery relies on existing information, while main effect filtering excludes SNPs whose effect can only be found when in combination with another SNP (i.e., SNPs that do not exhibit a main effect). In this study, main effect filtering yielded a larger number of significant SNP-SNP models. Caution is recommended when drawing conclusions, however, about the types of SNPs driving genetic interaction effects for the phenotypes in this study. Biofilter generated a larger number of models to test than the main effect filter, and as such, the smaller number of identified SNP-SNP models from Biofilter compared to the main effect filter may be due to the higher multiple testing burden.

Filtering methods allow us to overcome the multiple test burden and elucidate interactions among genes which are associated with complex diseases. Here, we exploit two filtering methods to identify G×G using GWAS and knowledge based Biofilter. Further incorporation of data from multi-omic databases, such as modifications of histones from the epigenome, would improve the understanding of complex traits and promote personalized interventions for adverse cardiac outcomes. Environmental factors and pharmaceutical elements could also be integrated to reveal complex gene-environment or gene-gene-environment interactions. Our findings demonstrate the power of evaluating G×G for cardiac-related health outcomes.

## Supporting information

**S1 Fig. Manhattan plots and QQ plots for 29 GWASs.** In the Manhattan plots, the blue line represents the genome wide significance ($-\log_{10}(5\times10^{-8}) = 7.30$) and the red line represents the suggestive Bonferroni significance ($-\log_{10}(0.05/577007) = 7.06$). Seven significant associations (at genome wide significance) between SNPs (rs174548, rs174549, rs4246215, rs174577, rs174583, rs174547, and rs174534) and log transformed Dihomo_g_Linolenic_C20_3n6 were identified. Two significant associations (at or close to the suggestive Bonferroni significance) were identified between rs11744802 and log transformed C18_2n6tt and between rs174577 and Arachidonic_acid_C20_4n6.
(PDF)

**S2 Fig.** Genetic principal component analysis (PCA) results: A) Distribution of the first 20 PCs and B) PC1 against PC2. The genetic principal components were calculated based on the individual genetic information. PC1 explained 0.103% of genetic variations within the LURIC cohort and PC2 could explained 0.0674% of genetic variation within the LURIC cohort. (PDF)

**S3 Fig.** Power simulations under the main effect filtering for number of SNP A) cardiac traits in logistic regression and B) fatty acids in linear regression, and for sample size with C) cardiac traits in logistic regression and D) fatty acids in linear regression. Power was estimated at varying beta coefficients under different regression models for sample size and for number of selected SNP for filtering. Each simulation was performed with 1000 replications. The desired power at 80% was labeled with black horizontal line and the sample size at 2824 or the number of selected SNP at 100 was labeled with red vertical. 80% power was reached with 100 SNPs (4,950 pairwise combinations of SNPs) if the beta coefficient is larger than 0.2 in the logistic regression (A) and if the beta coefficient is larger than 0.1 in the linear regression (B). 80% power was reached with 2824 people if the beta coefficient is larger than 0.2 in the logistic regression (C) and if the beta coefficient is larger than 0.1 in the linear regression (D). (PDF)

**S4 Fig.** Distribution of observed LRT p-values for the referential G×Gs based on the same set of 1000 random selected SNPs and G×G under two filtering methods to A) cardiac traits and B) fatty acids. G×G based on 1000 random selected SNPs was performed as baseline to understand the inflation (Black). G×G under main effect filtering (Blue) and G×G under Biofilter filtering (Red) were performed and compared with the G×G based on 1000 random selected SNPs for each phenotype. Individual Q-Q plots were generated to visualize the observed LRT p-values to the expected LRT p-values for cardiac traits and fatty acids. Red line represents the ideal estimation of LRT p-values. The corresponded genomic inflation factors were calculated for showing the model inflation (S2 Table). (PDF)

**S1 Table. Description for phenotypes in cardiac traits and fatty acids with post-QC sample size.**
(PDF)

**S2 Table.** Statistical summary of covariates and results of ANOVA test between covariates and A) cardiac traits or B) fatty acids. (PDF)

**S3 Table. Genomic inflation factor (Lambda) for G×G from the standard median approach and the linear regression approach for different models.** The SE median was reported for the standard median approach (M) and the SE was reported for the linear regression approach (R). (PDF)

**S4 Table. Summary for significant interactions among the cardiac traits and fatty acids under two filtering methods based on the PLATO output.** All significant gene-gene pairs for specific phenotypes were recorded with the SNP pairs, genes, p-values from GWAS, sample size, number of cases, beta coefficients, LRT p-values, FDR and Bonferroni-adjusted LRT p-values, and other information. Interactions for overall FDR and Bonferroni-adjusted LRT p-values less than 0.05 are highlighted in yellow, and interactions for only overall FDR-adjusted LRT p-value is less than 0.05 are highlighted in blue. The beginning of a new phenotype is highlighted in blue. Genes which were annotated by Biofilter with a 5kb boundary are

highlighted in brown. Genes which were annotated by Biofilter with a 5kb boundary are highlighted in brown.
(PDF)

**S1 File. Raw GWAS results for each phenotype.**
(PDF)

## Acknowledgments

We thank Xi He (The Pennsylvania State University) for development of the phenotype quality control pipeline, Anastasia M. Lucas (University of Pennsylvania) for examples of plotting the distribution of phenotype and genotype, Lin Song (University of Wisconsin-Madison) for proofreading the manuscript, and Mudong Zeng (The Pennsylvania State University) for consultation on statistical methodology and power simulations.

## Author Contributions

**Conceptualization:** Jiayan Zhou, Bertram Müller-Myhsok, Marcus E. Kleber, Winfried Maerz, Molly A. Hall.

**Data curation:** Kristin Passero, Nicole E. Palmiero, Bertram Müller-Myhsok, Marcus E. Kleber, Winfried Maerz, Molly A. Hall.

**Formal analysis:** Jiayan Zhou.

**Funding acquisition:** Bertram Müller-Myhsok, Marcus E. Kleber, Winfried Maerz, Molly A. Hall.

**Investigation:** Jiayan Zhou, Bertram Müller-Myhsok, Marcus E. Kleber, Winfried Maerz, Molly A. Hall.

**Methodology:** Jiayan Zhou, Kristin Passero, Nicole E. Palmiero, Bertram Müller-Myhsok, Molly A. Hall.

**Project administration:** Bertram Müller-Myhsok, Marcus E. Kleber, Winfried Maerz, Molly A. Hall.

**Resources:** Bertram Müller-Myhsok, Marcus E. Kleber, Winfried Maerz, Molly A. Hall.

**Software:** Jiayan Zhou, Kristin Passero, Nicole E. Palmiero.

**Supervision:** Bertram Müller-Myhsok, Marcus E. Kleber, Winfried Maerz, Molly A. Hall.

**Validation:** Jiayan Zhou, Kristin Passero, Nicole E. Palmiero.

**Visualization:** Jiayan Zhou, Bertram Müller-Myhsok.

**Writing – original draft:** Jiayan Zhou.

**Writing – review & editing:** Jiayan Zhou, Kristin Passero, Nicole E. Palmiero, Bertram Müller-Myhsok, Marcus E. Kleber, Molly A. Hall.

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
