## [Decision Letter · Decision Letter 0]

27 Mar 2020

PONE-D-19-31634

Investigation of gene-gene interactions in cardiac traits and serum fatty acid level in LURIC cohort

PLOS ONE

Dear Dr. Hall,

Thank you for submitting your manuscript to PLOS ONE. After careful consideration, we feel that it has merit but does not fully meet PLOS ONE’s publication criteria as it currently stands. Therefore, we invite you to submit a revised version of the manuscript that addresses the points raised during the review process.

We would appreciate receiving your revised manuscript by May 11 2020 11:59PM. To enhance the reproducibility of your results, we recommend that if applicable you deposit your laboratory protocols in protocols.io, where a protocol can be assigned its own identifier (DOI) such that it can be cited independently in the future. For instructions see: http://journals.plos.org/plosone/s/submission-guidelines#loc-laboratory-protocols

We look forward to receiving your revised manuscript.

Kind regards,

Heming Wang, PhD

Academic Editor

PLOS ONE

Journal Requirements:

3. Thank you for stating the following in the Competing Interests/Financial Disclosure* (delete as necessary) section:

"This work is supported by the USDA National Institute of Food and Agriculture and

Hatch Appropriations under Project #PEN04275 and Accession #1018544. The

funders had no role in study design, data collection and analysis, decision to publish,

or preparation of the manuscript.

Genotyping of the LURIC study participants was supported by the 7th Framework

Program AtheroRemo (grant agreement #201668) of the European Union. The funders

had no role in study design, data collection and analysis, decision to publish, or

preparation of the manuscript."

We note that one or more of the authors are employed by a commercial company: SyNergy and Synlab Holding Deutschland GmbH"

Additional Editor Comments (if provided):

Please provide more details about the main effect GWAS, including Manhattan plots, QQ plots, significance level for the top 100 SNPs etc in Supplements. Please also provide more background for Bilfilter analyses as reviewer 2 suggested.

The GxG analyses were performed on a relatively small samples without replication. It is hard to determine if those findings are true positive and can be generated to other populations. Can the authors look at other European ancestry studies for replication evidence? Please estimate statistical power for the GxG analyses.

Reviewers' comments:

Reviewer's Responses to Questions

**Comments to the Author**

1. Is the manuscript technically sound, and do the data support the conclusions?

Reviewer #1: Yes

Reviewer #2: Yes

2. Has the statistical analysis been performed appropriately and rigorously? 

Reviewer #1: Yes

Reviewer #2: Yes

3. Have the authors made all data underlying the findings in their manuscript fully available?

Reviewer #1: No

Reviewer #2: Yes

4. Is the manuscript presented in an intelligible fashion and written in standard English?

Reviewer #1: Yes

Reviewer #2: Yes

5. Review Comments to the Author

Reviewer #1: The authors applied two filtering methods to identify genetic interactions related to cardiac related outcomes from data from the Ludwigshafen Risk and Cardiovascular (LURIC) Health Study.

The manuscript appears technically sound and the the conclusions drawn are adequate.

As stated by the authors not all data underlying the findings in their manuscript are fully available.

The manuscript is concise, well structured and comprehensible.

Reviewer #2: The study by Zhou et al. identified the SNP–SNP epistatic interactions for a serial of cardiac traits and serum fatty acids in a cohort of 3,316 individuals. It is thought that gene-gene interaction can help to illustrate the genetic etiology of complex disease, but the epistasis identification is impeded by the computational and statistical burden from large numbers of SNP combinations that need to be tested. This study applied two filtering approaches before interaction testing, including main effect filtering based on GWAS and biological knowledge-based modeling through Biofilter. This manuscript is well written and the author clearly illustrated their methods and discussed the potential biological meanings of findings. However, I think the overall goal of this work is a little bit vague and the significances are not well presented. For example, what does the author want to highlight for this study? Is it about the method, or the new findings related to cardiac traits and serum acids?

Major Revisions:

1. Page 5, lines 94-96 in methods, the authors mentioned the covariates included in models. How does the author decide which covariate should be included? Were they just forced into the model? If so, were they all significant for every trait?

2. Page 7, lines 141-143 in methods, the authors mentioned the model for interaction testing also included the first three principal components. How did the author determine the number of PCs? How much variation was explained with the first three PCs?

3. Page 7, lines 152, “The top one hundred SNPs were selected from each phenotype’s GWAS.” Choose the top100 seems a little bit arbitrary. How was this selected? And did the author try to include more SNPs?

4. Page 7, lines 150-155, section “Main effect filtering and Biofilter filtering”. I would suggest the authors to describe more about how biofilter perform the interaction testing.

5. About methods, it would be much more clearly if the authors could add a simple diagram to illustrate the QC and analysis steps.

6. As indicated at the beginning, I think more attention should be paid to the biological implications of their findings. In regard to the significance of the findings, is there any replicates? Can these findings be validated in other cohorts? Additionally, were any of the identified interactions that were previously reported? The results would be more meaningful if author could address more about the biological significance of their findings by comparing to existing literatures.

7. Page 16, lines 330-332, “Notably, Biofilter-generated models were found to demonstrate larger interaction effect sizes than those found through main effect filtering…”. This is very interesting. Does the author find similar results in other literatures?

Minor Revision:

The p values can be showed with less digits.

6. PLOS authors have the option to publish the peer review history of their article (what does this mean?). If published, this will include your full peer review and any attached files.

Reviewer #1: No

Reviewer #2: No

---

## [Author Response · Author response to Decision Letter 0]

24 Jul 2020

RESPONSE TO REVIEWERS

Article Submitted to PLoS ONE

Manuscript Number: PONE-D-19-31634

Investigation of gene-gene interactions in cardiac traits and serum fatty acid level in LURIC cohort

We would like to thank the reviewers and editors for the time they spent reviewing our paper and the interest that was shown. Below are our detailed responses to reviewers’ comments which are in italics. All revisions in the manuscript have been highlighted in the tracked changes version of the submitted manuscript.

Additional Editor Comments:

Please provide more details about the main effect GWAS, including Manhattan plots, QQ plots, significance level for the top 100 SNPs etc in Supplements. Please also provide more background for Biofilter analyses as reviewer 2 suggested.

Thank you. To provide more details, we have added Manhattan plots from the GWAS, QQ plots, and also result files in the Supplement S1 Figure and Supplement S1 File. Additionally, detail was added to the Methods section.

For the main effect analysis, we performed GWAS for each of the phenotypes and selected the top 100 SNPs for each respective phenotype and performed subsequent comprehensive pairwise SNP-SNP interaction analyses for these SNP lists within each phenotype. Due to the fact that power for each phenotype will vary based on differing sample size, we wanted to give equal opportunity to each phenotype to identify genetic interactions. We performed simulation tests to identify an optimal SNP filter amount that yields 80% power for reasonable interaction effect sizes (S3 Fig A-B). Greater detail has been added to the Methods section of the manuscript. 

Finally, we added more details about the Biofilter analysis to the Methods section of the manuscript. Biofilter is a software used to build putative SNP-SNP pairs by referencing publicly available biological database knowledge, including published protein-protein interactions, pathways, and ontological information [1–3]. It has been successfully applied to numerous phenotypes, including cholesterol and lipid levels [1,4–6], efavirenz-containing HIV treatment [7], multiple-sclerosis [8], and cataracts [9,10], to identify genetic interactions through knowledge-based filtering, allowing the possibility of identifying SNPs that interact without demonstrating a main effect.

The GxG analyses were performed on a relatively small samples without replication. It is hard to determine if those findings are true positive and can be generated to other populations. Can the authors look at other European ancestry studies for replication evidence? Please estimate statistical power for the GxG analyses.

These are excellent points. Unfortunately, for this study we were restricted to only performing analysis on the LURIC cohort. However, we did estimate our power to detect replicating signals in another cohort through power calculations. We included our power simulation in Supplement S3 Figure C-D to show that our study was well powered to detect the gene-gene interactions with betas at 0.2 and higher for logistic model or betas at 0.1 and higher for linear model. We added descriptions of this in the Methods section of the manuscript. 

The number of published gene-gene interaction studies are limited for the phenotypes we examined in this study, and as such, we did not find any previously reported exact SNP-SNP interactions. However, we added further biological description regarding a gene-gene interaction we identified between XRCC4 and XRCC5 in the Results and Discussion sections of the manuscript. Further, we performed literature searches for the significant individual genes and their associated phenotypes to determine if any of the genes in the gene-gene interaction models had previous associations with the phenotypes and discussed examples of genes involved in multiple gene-gene interactions, including GCNT1, SCN2B, SCN4B, and ASTN2. This has been added to the Discussion section of the manuscript. 

Major Revisions:

1. Page 5, lines 94-96 in methods, the authors mentioned the covariates included in models. How does the author decide which covariate should be included? Were they just forced into the model? If so, were they all significant for every trait?

Response: Thank you. We chose to adjust for age, sex, waist-hip ratio, and BMI in this study because these variables have shown to be related to cardiac phenotypes [15]. We also performed ANOVA tests to show the relationships between each phenotype and covariate in S2 Table, which verified the need to adjust for these variables. This has been added to the Methods section of the manuscript.

2. Page 7, lines 141-143 in methods, the authors mentioned the model for interaction testing also included the first three principal components. How did the author determine the number of PCs? How much variation was explained with the first three PCs?

Response: We calculated all PCs based on individual genetic information. We plotted the distribution of the first 20 PCs and found that the first three PCs explained the majority of the variation (0.2324074%) (S2 Figure (A)). We also included a PCA plot of the samples based on the first two PCs (S2 Figure (B)). This information has been added to the Methods section of the manuscript.

3. Page 7, lines 152, “The top one hundred SNPs were selected from each phenotype’s GWAS.” Choose the top100 seems a little bit arbitrary. How was this selected? And did the author try to include more SNPs?

Response: As mentioned above, due to the fact that power for each phenotype will vary based on differing sample size, we wanted to give equal opportunity to each phenotype to identify genetic interactions and so we chose to use the top 100 SNPs as a filter rather than a p-value cutoff as the number of qualifying SNPs for this approach will widely vary across the phenotypes. Further, we estimated the power (S3 Figure (C and D)) prior to implementing our analysis plan and chose to use top 100 as the optimal filtering cutoff. 

4. Page 7, lines 150-155, section “Main effect filtering and Biofilter filtering”. I would suggest the authors to describe more about how biofilter perform the interaction testing.

Response: Thank you for this comment. We have made changes to add more detail, as described in the response to first comment from the editor above.

5. About methods, it would be much more clearly if the authors could add a simple diagram to illustrate the QC and analysis steps.

Response: We found this to be a very helpful suggestion. We added a flowchart to show the genotype QC, phenotype QC, and also the interactions to the manuscript (Figure 1).

6. As indicated at the beginning, I think more attention should be paid to the biological implications of their findings. In regard to the significance of the findings, is there any replicates? Can these findings be validated in other cohorts? Additionally, were any of the identified interactions that were previously reported? The results would be more meaningful if author could address more about the biological significance of their findings by comparing to existing literatures.

Response: Thank you for suggesting this. As only 2,824 people were available after the QC and we had to ensure power to detect interaction signal, we performed a discovery study with the LURIC cohort without replication. As described in the response to the second Editor Comment, we have provided power estimates in S3 Figure that indicate the sample size and number of SNPs needed for replication in future studies. 

As described in the second Editor Comment, we searched all identified interactions in the gene knowledge databases and also the significant individual genes and their associated phenotypes to determine if any of the genes in the genetic models had previous associations. These details have been added to the Discussion section of the manuscript.

7. Page 16, lines 330-332, “Notably, Biofilter-generated models were found to demonstrate larger interaction effect sizes than those found through main effect filtering…”. This is very interesting. Does the author find similar results in other literatures?

Response: Thank you very much for this comment; this finding has not been looked into previously, and we appreciate your interest. Upon further investigation of previously published work involving both biofilter and main effect filtering for gene-gene interaction analysis, we found other examples in which the biofilter approach yielded larger interaction effect sizes than main effect filtering, for example, the gene-gene interaction analyses of lipids [5] and type II diabetes [16]. We do not know the reason for this finding at this time, and we would like to investigate this in future research. This has been added to the Discussion section of the manuscript.

Minor Revision:

The p values can be showed with less digits.

Response: We have removed additional digits for the p-values to include only four digits. 

 

Reference

1. Kim D, Lucas A, Glessner J, Verma SS, Bradford Y, Li R, et al. BIOFILTER AS A FUNCTIONAL ANNOTATION PIPELINE FOR COMMON AND RARE COPY NUMBER BURDEN. Pac Symp Biocomput. 2016;21: 357–68. Available: http://www.ncbi.nlm.nih.gov/pubmed/26776200

2. Pendergrass SA, Frase A, Wallace J, Wolfe D, Katiyar N, Moore C, et al. Genomic analyses with biofilter 2.0: Knowledge driven filtering, annotation, and model development. BioData Min. 2013;6: 25. doi:10.1186/1756-0381-6-25

3. Bush WS, Dudek SM, Ritchie MD. Biofilter: A knowledge-integration system for the multi-locus analysis of genome-wide association studies. Pacific Symposium on Biocomputing 2009, PSB 2009. NIH Public Access; 2009. pp. 368–379. 

4. Turner SD, Berg RL, Linneman JG, Peissig PL, Crawford DC, Denny JC, et al. Knowledge-driven multi-locus analysis reveals gene-gene interactions influencing HDL cholesterol level in two independent EMR-linked biobanks. PLoS One. 2011;6: e19586. doi:10.1371/journal.pone.0019586

5. Holzinger ER, Verma SS, Moore CB, Hall M, De R, Gilbert-Diamond D, et al. Discovery and replication of SNP-SNP interactions for quantitative lipid traits in over 60,000 individuals. BioData Min. 2017;10: 25. doi:10.1186/s13040-017-0145-5

6. De R, Verma SS, Holzinger E, Hall M, Burt A, Carrell DS, et al. Identifying gene–gene interactions that are highly associated with four quantitative lipid traits across multiple cohorts. Hum Genet. 2017;136: 165–178. doi:10.1007/s00439-016-1738-7

7. Grady BJ, Torstenson ES, McLaren PJ, De Bakker PIW, Haas DW, Robbins GK, et al. Use of biological knowledge to inform the analysis of gene-gene interactions involved in modulating virologic failure with efavirenz-containing treatment regimens in art-naïve ACTG clinical trials participants. Pacific Symposium on Biocomputing 2011, PSB 2011. 2011. pp. 253–264. 

8. Bush WS, McCauley JL, Dejager PL, Dudek SM, Hafler DA, Gibson RA, et al. A knowledge-driven interaction analysis reveals potential neurodegenerative mechanism of multiple sclerosis susceptibility. Genes Immun. 2011;12: 335–340. doi:10.1038/gene.2011.3

9. Pendergrass SA, Verma SS, Hall MA, Holzinger ER, Moore CB, Wallace JR, et al. Next-generation analysis of cataracts: determining knowledge driven gene-gene interactions using biofilter, and gene-environment interactions using the Phenx Toolkit*. Pac Symp Biocomput. 2015; 495–505. 

10. Hall MA, Verma SS, Wallace J, Lucas A, Berg RL, Connolly J, et al. Biology-Driven Gene-Gene Interaction Analysis of Age-Related Cataract in the eMERGE Network. Genet Epidemiol. 2015;39: 376–384. doi:10.1002/gepi.21902

11. Zhang Q, Karnak D, Tan M, Lawrence TS, Morgan MA, Sun Y. FBXW7 Facilitates Nonhomologous End-Joining via K63-Linked Polyubiquitylation of XRCC4. Mol Cell. 2016;61: 419–433. doi:10.1016/j.molcel.2015.12.010

12. Drouet J, Delteil C, Lefrançois J, Concannon P, Salles B, Calsou P. DNA-dependent protein kinase and XRCC4-DNA ligase IV mobilization in the cell in response to DNA double strand breaks. J Biol Chem. 2005;280: 7060–7069. doi:10.1074/jbc.M410746200

13. Gene [Internet]. Bethesda (MD): National Library of Medicine (US), National Center for Biotechnology Information; Accession No. 7518, X-ray repair cross complementing 4 (XRCC4). 2004 [cited 21 Feb 2019]. Available: https://www.ncbi.nlm.nih.gov/gene/7518

14. Gene [Internet]. Bethesda (MD): National Library of Medicine (US), National Center for Biotechnology Information; Accession No. 7520, X-ray repair cross complementing 5 (XRCC5). 2004 [cited 21 Feb 2019]. Available: https://www.ncbi.nlm.nih.gov/gene/7520

15. CDC. Heart Disease Facts | cdc.gov. In: Center for Disease Control [Internet]. 2020 [cited 7 Apr 2020]. Available: https://www.cdc.gov/heartdisease/facts.htm

16. Hall MA, Wallace J, Lucas A, Kim D, Basile AO, Verma SS, et al. PLATO software provides analytic framework for investigating complexity beyond genome-wide association studies. Nat Commun. 2017;8: 1167. doi:10.1038/s41467-017-00802-2

---

## [Decision Letter · Decision Letter 1]

14 Aug 2020

Investigation of gene-gene interactions in cardiac traits and serum fatty acid levels in LURIC cohort

PONE-D-19-31634R1

Dear Dr. Hall,

We’re pleased to inform you that your manuscript has been judged scientifically suitable for publication and will be formally accepted for publication once it meets all outstanding technical requirements.

Kind regards,

Heming Wang, PhD

Academic Editor

PLOS ONE

Additional Editor Comments (optional):

Reviewers' comments:

Reviewer's Responses to Questions

**Comments to the Author**

1. If the authors have adequately addressed your comments raised in a previous round of review and you feel that this manuscript is now acceptable for publication, you may indicate that here to bypass the “Comments to the Author” section, enter your conflict of interest statement in the “Confidential to Editor” section, and submit your "Accept" recommendation.

Reviewer #2: All comments have been addressed

2. Is the manuscript technically sound, and do the data support the conclusions?

Reviewer #2: Yes

3. Has the statistical analysis been performed appropriately and rigorously? 

Reviewer #2: Yes

4. Have the authors made all data underlying the findings in their manuscript fully available?

Reviewer #2: Yes

5. Is the manuscript presented in an intelligible fashion and written in standard English?

Reviewer #2: Yes

6. Review Comments to the Author

Reviewer #2: The author has fully addressed my concerns with more illustrations in Methods and Discussion. I don't have any additional comments.

7. PLOS authors have the option to publish the peer review history of their article (what does this mean?). If published, this will include your full peer review and any attached files.

Reviewer #2: **Yes: **Jingting Yu

---

## [Editor Report · Acceptance letter]

24 Aug 2020

PONE-D-19-31634R1 

Investigation of gene-gene interactions in cardiac traits and serum fatty acid levels in LURIC cohort 

Dear Dr. Hall:

I'm pleased to inform you that your manuscript has been deemed suitable for publication in PLOS ONE. Congratulations! Your manuscript is now with our production department. 

Kind regards, 

on behalf of

Dr. Heming Wang 

Academic Editor

PLOS ONE